# Mitigating gut microbial degradation of levodopa and enhancing brain dopamine: Implications in Parkinson's disease

Gang Cheng[1], Micael Hardy [2], Cecilia J. Hillard [3], Jimmy B. Feix[1] & Balaraman Kalyanaraman [1] ✉

Parkinson's disease is managed using levodopa; however, as Parkinson's disease progresses, patients require increased doses of levodopa, which can cause undesirable side effects. Additionally, the oral bioavailability of levodopa decreases in Parkinson's disease patients due to the increased metabolism of levodopa to dopamine by gut bacteria, *Enterococcus faecalis*, resulting in decreased neuronal uptake and dopamine formation. Parkinson's disease patients have varying levels of these bacteria. Thus, decreasing bacterial metabolism is a promising therapeutic approach to enhance the bioavailability of levodopa in the brain. In this work, we show that Mito-*ortho*-HNK, formed by modification of a naturally occurring molecule, honokiol, conjugated to a triphenylphosphonium moiety, mitigates the metabolism of levodopa—alone or combined with carbidopa—to dopamine. Mito-*ortho*-HNK suppresses the growth of *E. faecalis*, decreases dopamine levels in the gut, and increases dopamine levels in the brain. Mitigating the gut bacterial metabolism of levodopa as shown here could enhance its efficacy.

For decades, levodopa (L-dopa) has been used to manage symptoms caused by the depletion of endogenous dopamine in the brains of patients with Parkinson's disease (PD)[1]. A characteristic hallmark of PD is the loss of dopamine-producing neurons in the striatum. Orally administered L-dopa crosses the blood–brain barrier and is metabolized to dopamine in the brain by the enzyme aromatic L-amino acid decarboxylase. As PD progresses, patients require increased doses of L-dopa, the side effects of which are dyskinesia and drug toxicity[1]. Unfortunately, as PD progresses, more dopaminergic neurons are lost, which—together with likely development of tolerance to dopamine effects at its receptors—results in an increased need for L-dopa to maintain therapeutic efficacy. As L-dopa doses are increased, however, the potential for the development of L-dopa induced dyskinesia is enhanced. L-dopa induced dyskinesia is extremely debilitating and often requires withdrawal of L-dopa therapy[2–4].

Nearly five decades ago, it was reported that L-dopa can be metabolized by the gut microbiota[5]. Therapeutic implications of extracerebral metabolism of L-dopa in treatment of PD were also discussed. Only recently, the specific genes and enzymes of the microbial metabolism of L-dopa were identified[6–8]. The decrease in the oral bioavailability of L-dopa in patients with PD was linked to its increased metabolism to dopamine by gut bacteria, resulting in decreased uptake of L-dopa and

dopamine formation in the brain[6–8]. Dopamine produced by decarboxylation of L-dopa in the gut does not cross the blood–brain barrier. In particular, the presence of commensal *Enterococcus faecalis* in the gut microbiota of PD patients correlates with increased metabolism of L-dopa in gut microbiota and decreased striatal dopamine levels[7,8]. Thus, decreasing the microbial metabolism of L-dopa is a potentially promising approach to enhancing its bioavailability and, as a result, its conversion to dopamine in the brain.

Using genome-mining techniques, *E. faecalis* was identified as the microbial species responsible for L-dopa metabolism[5]. Investigators discovered that L-dopa is metabolized to dopamine by *E. faecalis*-derived tyrosine decarboxylases[6–8]. Ex vivo human fecal suspensions from PD patients and healthy individuals were used to show that *E. faecalis* in gut microbiota is responsible for L-dopa metabolism[6–8]. Thus, L-dopa metabolism by gut commensal bacteria results in reduced therapeutic efficacy of L-dopa as well as increased *m*-tyramine generation from dopamine, which can have serious adverse effects[6–8]. Alpha-fluoromethyl amino acids, known inhibitors of tyrosine decarboxylases, and the L-tyrosine analog (S)-alpha-fluoromethyltyrosine, both inhibit L-dopa decarboxylation in *E. faecalis*[7]. The role of the microbiota–gut–brain axis in regulating dopaminergic signaling is gaining increased attention[9].

[1]Department of Biophysics, Medical College of Wisconsin, 8701 Watertown Plank Road, Milwaukee, WI 53226, USA. [2]Aix-Marseille Univ, CNRS, ICR, UMR 7273, Marseille 13013, France. [3]Department of Pharmacology and Toxicology and Neuroscience Research Center, Medical College of Wisconsin, 8701 Watertown Plank Road, Milwaukee, WI 53226, USA. ✉e-mail: balarama@mcw.edu

Bacterial tyrosine decarboxylases are present in all but three of 655 *E. faecalis* genomes surveyed[6]. Typically, carbidopa, an inhibitor of extra-cerebral aromatic L-amino acid decarboxylase, prevents its peripheral metabolism to dopamine[10]. However, carbidopa is not an effective inhibitor of the enzyme tyrosine decarboxylases; it is 200-fold less active toward *E. faecalis* tyrosine decarboxylases relative to human dopa decarboxylase and is unable to prevent gut bacterial L-dopa metabolism[6–8].

Antimicrobial therapy has been suggested as a nontoxic, highly effective approach to mitigating the gut metabolism of L-dopa to dopamine and enhancing the uptake and metabolism of L-dopa into the brain in murine models of PD[11]. Cationic compounds can interact with bacterial membranes and inhibit bacterial growth[12–14], including triphenylphosphonium cation (TPP$^+$)-based mitochondria-targeted compounds that have been shown to be effective antimicrobial agents[15,16]. Compounds that decrease the electrochemical gradient of protons across the bacterial cytoplasmic membrane (also known as proton motive force) were reported to exert antibacterial effects[16]. Compounds containing the hydrophobic cations such as TPP$^+$ with a delocalized positive charge exert protonophore-like activity in mitochondrial membranes[17]. Although several TPP$^+$-based mitochondria-targeted drugs (MTDs) exert antimicrobial effects[16], we chose to investigate the effect of mitochondria-targeted honokiol (Mito-*ortho*-HNK) and analogs on bacterial metabolism of L-dopa because these compounds have previously been used in preclinical animal models without significant toxicity[18,19].

Results from our studies show that Mito-*ortho*-HNK suppressed the growth of *E. faecalis* in a time- and dose-dependent manner. Mito-HNK decreased levodopa degradation and dopamine formation by *E. faecalis*. Additionally, results show that administration of Mito-*ortho*-HNK along with deuterated L-dopa to mice decreased deuterated dopamine formation in the gut and enhanced the uptake of deuterated L-dopa and formation of deuterated dopamine in the brain. These findings suggest a potential therapeutic pathway for enhancing the efficacy of L-dopa and L-dopa/carbidopa therapy in PD.

## Results

### Mito-*ortho*-HNK and Mito-PEG$_4$-HNK and analogs inhibit *E. faecalis* proliferation

Mito-*ortho*-HNK and PEGylated mitochondria-targeted honokiol (Mito-PEG$_4$-HNK) are nearly 20 times more effective than unmodified honokiol (HNK) in inhibiting *E. faecalis* proliferation (Fig. 1a, c). At low concentrations, Mito-*ortho*-HNK analogs delayed the proliferation of *E. faecalis*. At lower doses of Mito-*ortho*-HNK and analogs, after a 2–3 h time lag, *E. faecalis* proliferation continued at nearly the same rate as the control. (Fig. 1b, d).

### Mito-*ortho*-HNK and analogs inhibit L-dopa metabolism to dopamine by *E. faecalis*

Next, we performed similar experiments in the presence of L-dopa. As shown in Fig. 2a, Mito-*ortho*-HNK dose-dependently inhibited *E. faecalis* proliferation in the presence of L-dopa (1 mM). Samples collected at different time points were analyzed by high-performance liquid chromatography (Fig. 2b). At concentrations of 3 µM and 4 µM of Mito-*ortho*-HNK, L-dopa degradation was inhibited together with reduced dopamine formation by *E. faecalis* (Fig. 2c, d). Samples collected at the 4 h time point showed that treatment with 3 µM of Mito-*ortho*-HNK caused 71% and 68% inhibition in L-dopa consumption and dopamine formation, respectively, compared with the control group in the same period.

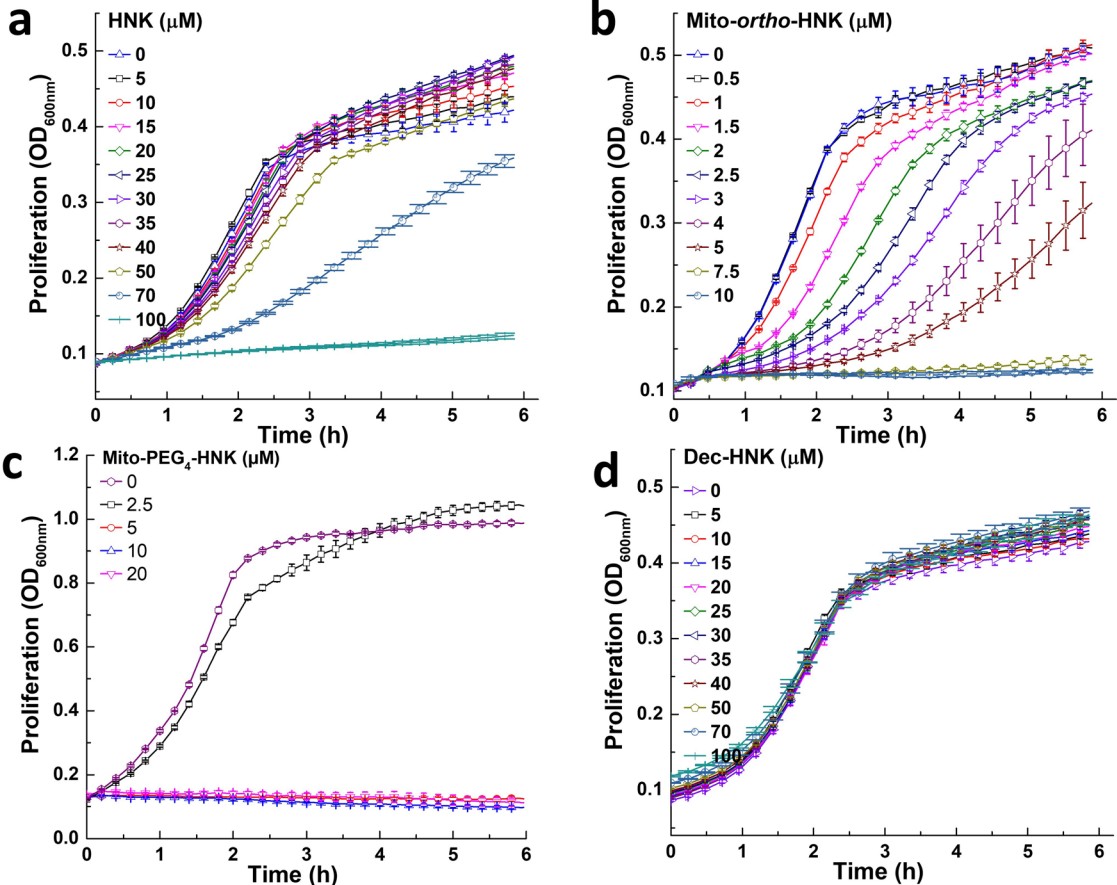

**Fig. 1 | The effects of Mito-*ortho*-HNK analogs on the bacterial proliferation of *E. faecalis*.** The effects of HNK (**a**), Mito-*ortho*-HNK (**b**), Mito-PEG$_4$-HNK (**c**), Dec-HNK (**d**) on the proliferation of *E. faecalis* were monitored at OD$_{600}$ for 6 h. Data shown are the mean ± SD, *n* = 4.

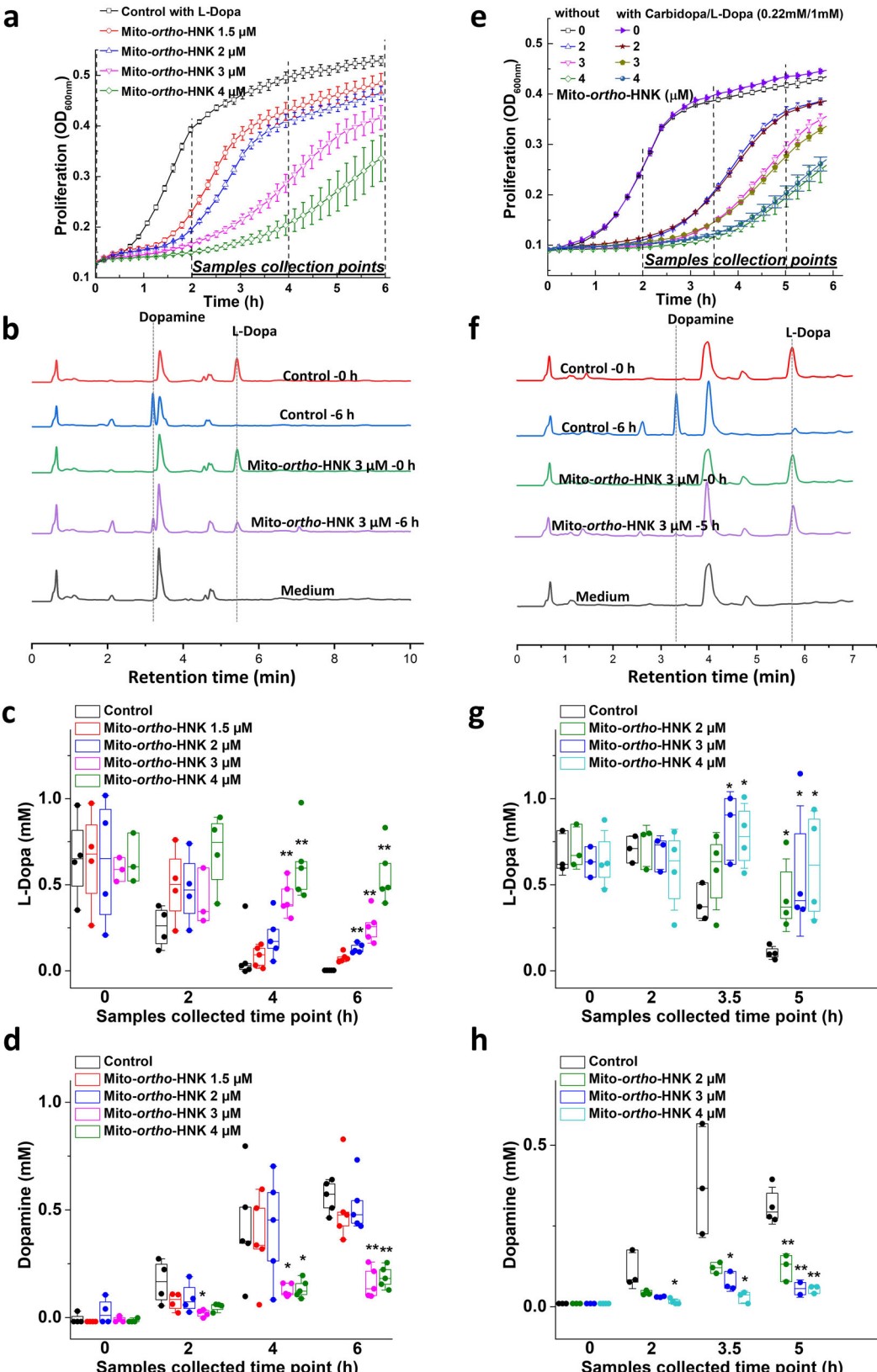

**Fig. 2 | Effects of Mito-*ortho*-HNK on the bacterial L-dopa consumptions with or without carbidopa.** *E. faecalis* was treated with the Mito-*ortho*-HNK as indicated in the presence of 1 mM L-dopa alone (**a–d**) or with the additional 0.22 mM of carbidopa (**e–h**). The effects of Mito-*ortho*-HNK on the proliferation in the presence of L-dopa alone (**a**) or in the presence of L-dopa and carbidopa combination (**e**) were monitored at OD$_{600}$ for 6 h and culture media were collected at indicated time points for L-dopa and dopamine measurements. High-performance liquid

chromatography traces of representatives' samples and standards are shown in panel (**b**, **f**). The effects of Mito-*ortho*-HNK on L-dopa consumption (**c**, **g**) and dopamine formation (**d**, **h**) are shown in the presence of 1 mM L-dopa alone (**a–d**) or with the additional 0.22 mM of carbidopa (**e–h**) respectively. Statistical significance was determined using a Student's *t* test. *$P < 0.05$ or **$P < 0.01$ vs control group at each collection time point. Data shown are the mean ± SD, $n = 3$.

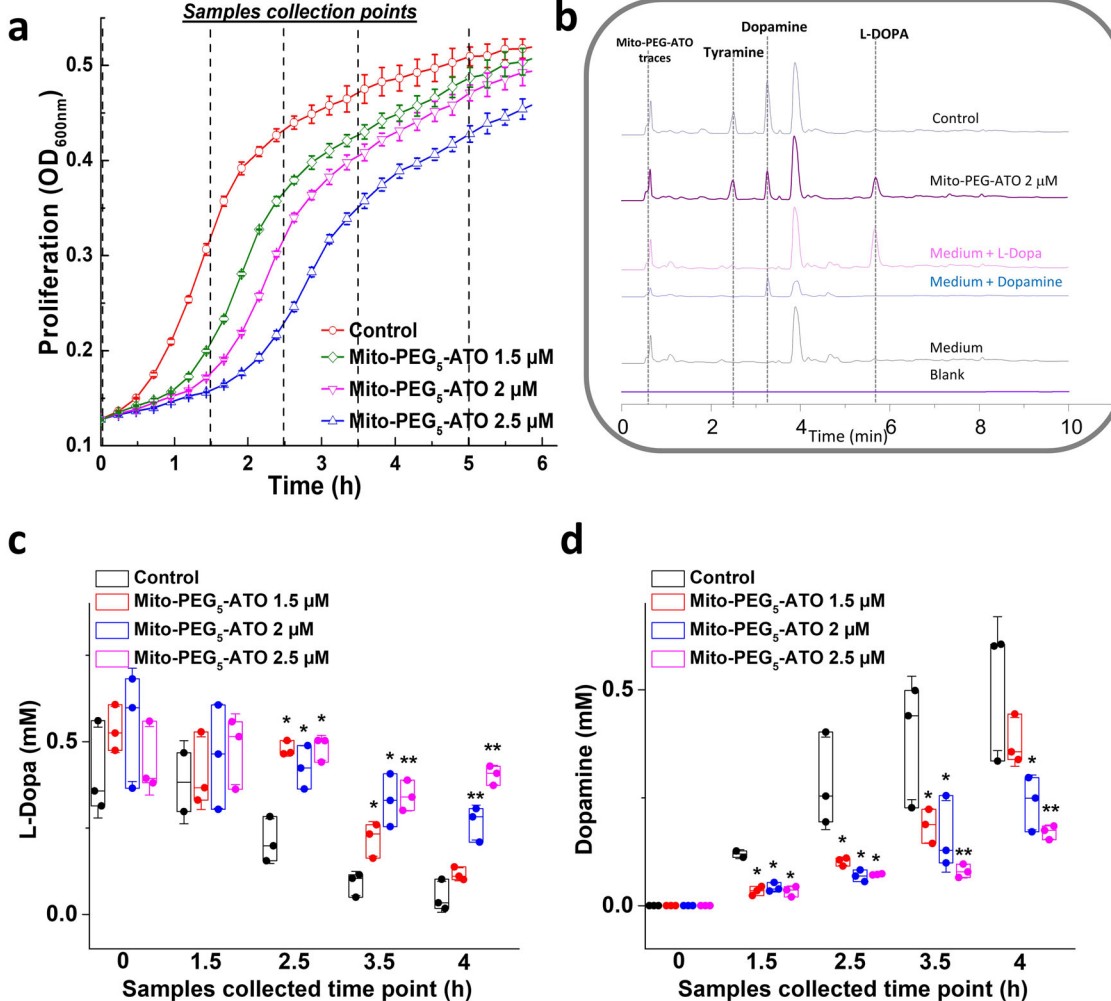

**Fig. 3 | Effects of Mito-PEG5-ATO on the bacterial L-dopa consumptions.**
*E. faecalis* was treated with Mito-PEG5-ATO as indicated in presence of L-dopa
(1 mM). The effects of Mito-PEG5-ATO on the proliferation (**a**) were monitored at
OD600 for 6 h and culture media were collected at indicated time points for L-dopa
and dopamine measurements. High-performance liquid chromatography traces of

representatives' samples and standards are shown in (**b**). The effects of Mito-PEG5-
ATO on L-dopa consumption (**c**) and dopamine formation (**d**) were shown in panels
(**c**) and (**d**), respectively. Statistical significance was determined using a Student's *t*
test. *$P < 0.05$ or **$P < 0.01$ vs control group at each collection time point. Data
shown are the mean ± SD, $n = 3$.

L-dopa is routinely used in combination with carbidopa in PD man-
agement. We investigated the effect of carbidopa (using the same ratio of
L-dopa:carbidopa as used in clinic) on L-dopa metabolism by *E. faecalis* and
the effect of Mito-*ortho*-HNK. As shown in Fig. 2e–h, Mito-*ortho*-HNK
dose-dependently induced a time lag in *E. faecalis* proliferation in the pre-
sence of L-dopa (1 mM) and carbidopa (0.2 mM). Samples collected at
different points were analyzed by high-performance liquid chromatography
(Fig. 2e–h). At the concentration used, carbidopa did not affect the L-dopa
metabolism as it is a poor substrate for bacterial tyrosine decarboxylase[6-8].

### Mito-PEG5-ATO inhibits L-dopa metabolism to dopamine by *E. faecalis*

As shown in Fig. 3a, PEGylated mitochondria-targeted atovaquone (Mito-
PEG5-ATO) dose-dependently inhibited *E. faecalis* proliferation in the
presence of L-dopa (1 mM). Samples isolated at various time points were
analyzed by high-performance liquid chromatography (Fig. 3b). Mito-
PEG5-ATO dose dependently inhibited L-dopa consumption (Fig. 3c) by *E.
faecalis* and the formation of dopamine (Fig. 3c). Samples collected at the
2.5 h time point showed that treatment with 2.5 μM Mito-PEG5-ATO
caused a 50% inhibition in L-dopa consumption and dopamine formation
(Fig. 3c, d).

### Antimicrobial effects of MTDs on gut bacteria

To determine the impact of MTDs and related controls on bacterial growth,
the minimal inhibitory concentration values were measured by micro
broth-dilution assay[20]. Minimal inhibitory concentration is the lowest
concentration of an antibiotic that fully inhibits the growth of a bacterial
strain. As shown in Table 1, all of the compounds were antimicrobial against
a panel of representative gram-positive bacteria, with minimal inhibitory
concentration values in the 0.5–4 μM range. Antimicrobial activity against
two representative gram-negative bacteria, *Escherichia coli* (BL21) and
*Pseudomonas aeruginosa* (PA01), was relatively weaker, with minimal
inhibitory concentration values from 16 μM to >64 μM (Table 1). This may
be due to several factors including reduced uptake across the more complex
gram-negative outer membrane and the presence of a multidrug resistance
pump[16].

### Comparison of antimicrobial effects of commonly used antibiotics and MTDs

Preliminary results also indicated that commonly used antibiotics chlor-
amphenicol (1.25–20 μM) and ampicillin (2.5–20 μM) inhibited *E. faecalis*
proliferation throughout the 6 h monitoring period (Fig. 4). However, Mito-
*ortho*-HNK and analogs exhibited a time lag depending on the

**Table 1 | Effect of mitochondria-targeted analogs on minimal inhibitory concentrations in gram negative and positive bacteria**

| | Minimal inhibitory concentrations (µM) | | | | | |
|---|---|---|---|---|---|---|
| | *S. aureus* | *B. subtilis* | *E. faecium* | *E. faecalis* | *E. coli* BL21 | *P. aerug* PA01 |
| HNK | 32 | 64 | 64 | 64 | NA | NA |
| Mito-*ortho*-HNK | 1 | 1 | 0.5 | 2 | NA | NA |
| Decyl-HNK | NA | NA | NA | NA | 16 | 32 |
| Mito$_{10}$-ATO | 16 | 4 | 8 | 8 | 64 | 64 |
| Mito-PEG$_2$-ATO | 1 | 1 | 1 | 1 | 4 | 8 |
| Mito-PEG$_4$-ATO | 1 | 1 | 1 | 1 | 4 | 8 |
| Mito-PEG$_5$-ATO | 1 | 1 | 1 | 1 | 8 | 16 |
| Mito-PEG$_9$-ATO | 2 | 2 | 2 | 4 | 8 | 16 |

No inhibition of cell growth at 64 µM.

*NA* not active.

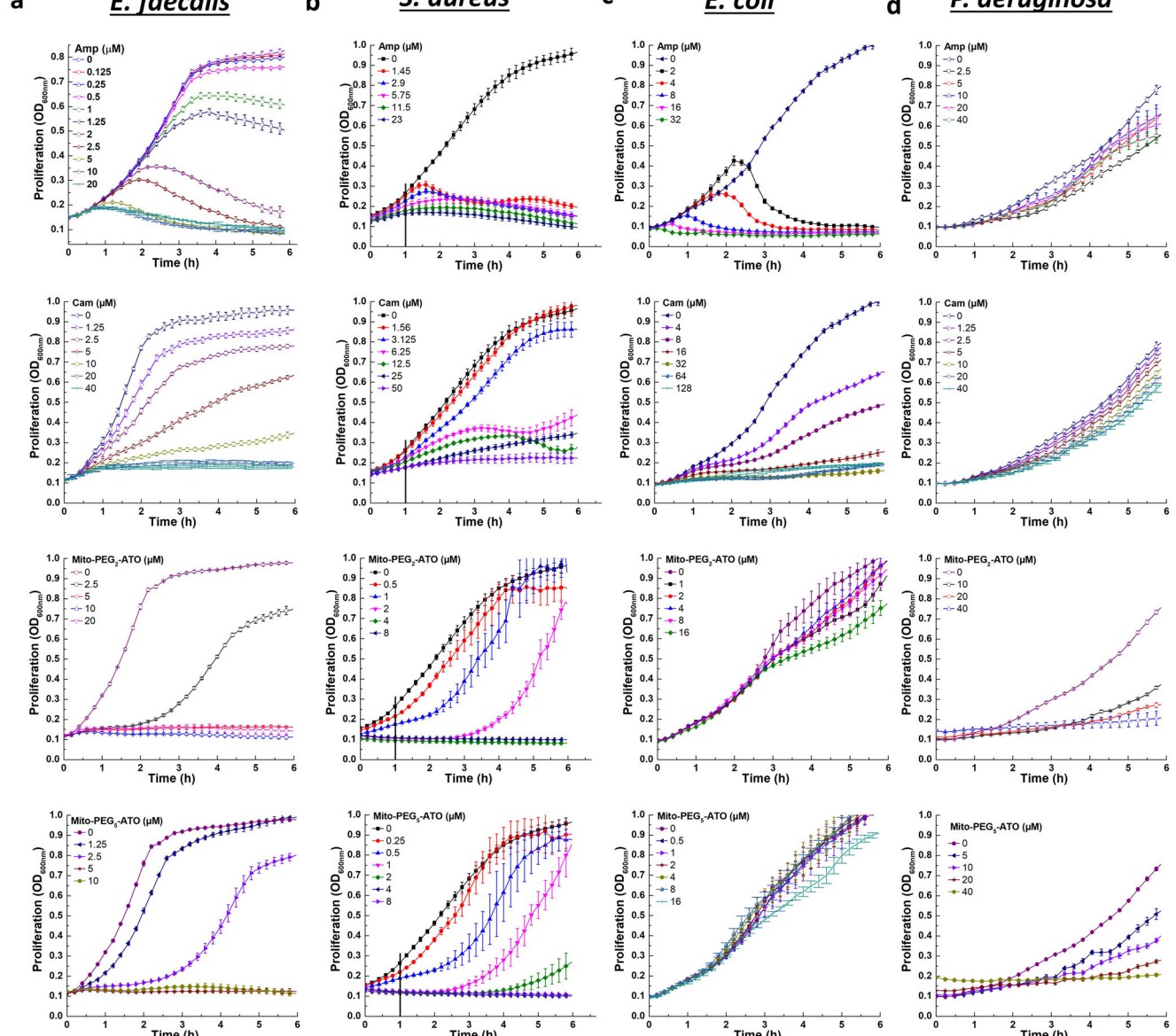

**Fig. 4 | The effects of commonly used antibiotics and Mito-PEG-ATO analogs on the proliferation of gram-positive and gram-negative bacteria.** The effects of commonly used antibiotics (chloramphenicol [Cam] and ampicillin [Amp]), Mito-PEG$_2$-ATO, and Mito-PEG$_5$-ATO on the proliferation were monitored at OD$_{600}$ for 6 h in gram-positive *E. faecalis* (**a**) and *S. aureus* (**b**) and in gram-negative *E. coli* (**c**) and *P. aeruginosa* (**d**). Data shown are the mean ± SD, *n* = 4.

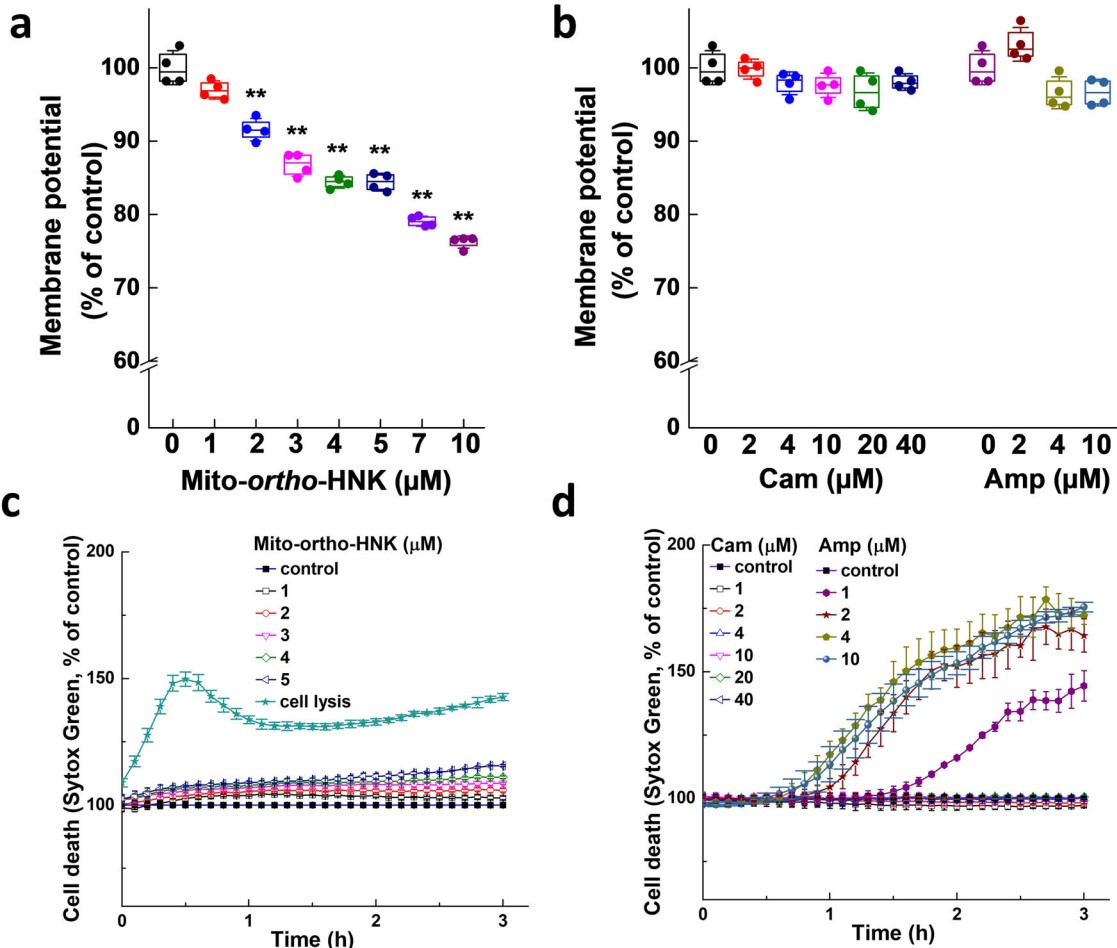

**Fig. 5 | Effects of Mito-*ortho*-HNK and commonly used antibiotics on the bacterial membrane potential. a**, **b** *E. faecalis* was treated with Mito-*ortho*-HNK (**a**) or commonly used antibiotics (**b**), chloramphenicol [Cam] and ampicillin [Amp]) as indicated for 1 h. The effects on the membrane potential were measured by TMRM dye (50 nM), the fluorescence indicator, to determine the percentage change in TMRM fluorescence intensity between the control and treatments groups. The lower levels of TMRM fluorescence resulting from treatment reflect the depolarization of mitochondrial membrane potential. Statistical significance was determined using a Student's *t* test. **$P < 0.01$ vs control group at each collection time point. Data shown are the mean ± SD, $n = 4$. **c**, **d** *E. faecalis* was treated with Mito-*ortho*-HNK (**c**) or commonly used antibiotics (**d**) as indicated for 3 h, and cell death was monitored in real time by SYTOX Green staining. Data shown are the mean ± SD for $n = 4$.

concentration. At 2.5 µM concentration, *E. faecalis* proliferation was maximally inhibited for 2 h, and then recovered to reach a similar level of confluence as the control (Fig. 2).

The effects of commonly used antibiotics and Mito-PEG-ATO analogs on the proliferation of *E. faecalis* are shown in Fig. 4. Results show that chloramphenicol and ampicillin dose-dependently inhibited *E. faecalis* proliferation. However, Mito-PEG-ATO and analogs exhibited a time lag depending on the concentration. At 2.5 µM concentration, *E. faecalis* proliferation was maximally inhibited for 2 h, and then it started to proliferate and reached a similar level of confluence as control.

This raises the possibility that L-dopa metabolism by *E. faecalis* can be inhibited in the presence of MTDs, including Mito-*ortho*-HNK and analogs or Mito-PEG-ATO analogs, and that the extent of inhibition is dependent on the timing of administration. If both L-dopa and Mito-*ortho*-HNK analogs were added at the same time, it is suggested that L-dopa metabolism to dopamine would be inhibited during the initial time lag and would resume with time.

### Effect of MTDs on membrane potential
The uptake of TPP$^+$-modified compounds into bacteria is driven by the large, negative inside potential across the bacterial envelope[12,16]. Membrane depolarization was observed for the MTD SkQ1 and proposed as its mechanism of action[12]. Fig. 5a shows that Mito-*ortho*-HNK dose-dependently decreased tetramethylrhodamine (TMRM) fluorescence due to depolarization of bacterial membrane potential. Treatment with a 4 µM concentration of Mito-*ortho*-HNK decreased the membrane potential by 15%. In contrast with MTDs, treatment with commonly used antibiotics chloramphenicol and ampicillin had no effect on TMRM fluorescence (Fig. 5b).

### Effect of Mito-*ortho*-HNK and commonly used antibiotics on cytotoxicity
Cytotoxicity was monitored in real time by SYTOX Green staining. The SYTOX measurements showed that Mito-*ortho*-HNK was not cytotoxic at the concentration inhibiting >90% proliferation of *E. faecalis* (Fig. 5c). In contrast, the antibiotic ampicillin but not chloramphenicol was cytotoxic at concentrations inhibiting *E. faecalis* proliferation (Fig. 5d). The Discussion section explores the mechanistic differences.

### Effect of Mito-*ortho*-HNK and commonly used antibiotics on ATP
Intracellular ATP was measured in *E. faecalis* cells in the presence of Mito-*ortho*-HNK and the commonly used antibiotics chloramphenicol and ampicillin. As shown in Fig. 6, the effects of Mito-*ortho*-HNK and antibiotics on ATP formation were completely different. Whereas Mito-*ortho*-HNK induced a dose- and time-dependent increase in ATP, ampicillin caused a dose- and time-dependent decrease in ATP compared with the

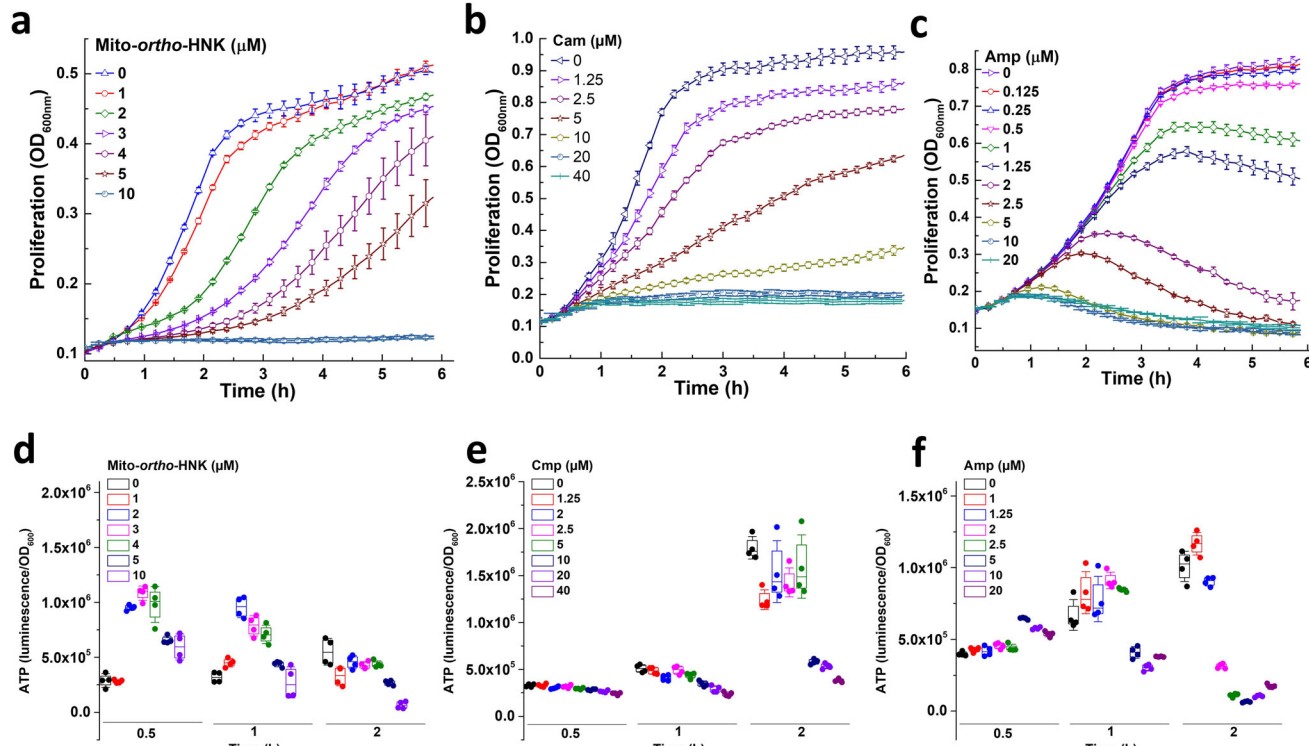

**Fig. 6 | Effects of Mito-*ortho*-HNK and antibiotics on ATP.** E. *faecalis* were treated with Mito-*ortho*-HNK (**a**) or commonly used antibiotics (chloramphenicol [Cam], (**b**); ampicillin [Amp], **c**) as indicated. Panels (**a–c**) show the effects of Mito-*ortho*-HNK or antibiotics on E. *faecalis* proliferation shown as absorbance at $OD_{600}$. Panels (**d–f**) show the effects of Mito-*ortho*-HNK or antibiotics on intracellular ATP levels as total luciferin luminescence. Data shown are the mean ± SD, $n = 4$.

control in an E. *faecalis* system. As shown in the proliferation data (Fig. 6a), in Mito-*ortho*-HNK-treated E. *faecalis*, ATP levels were higher, probably due to decreased utilization of ATP under suppressed proliferation. After a dose-dependent time lag, the rates of proliferation of E. *faecalis* increased with time. Concomitantly, ATP levels started to decrease due to increased utilization of ATP (Fig. 6d–f). In contrast with Mito-*ortho*-HNK, commonly used antibiotics chloramphenicol and ampicillin did not increase ATP levels in E. *faecalis* compared with the control. At higher concentrations, ATP levels decreased below the control levels, despite a decreased rate of proliferation.

### Effect of MTD on in vivo metabolism ʟ-dopa

The basal level of dopamine in the mouse brain is high. In order to monitor the impact of MTD treatment on ʟ-dopa uptake, we used levodopa-d3 (ʟ-dopa-d3), a stable form of ʟ-dopa with a deuterium substituted aromatic ring that crosses the blood–brain barrier (Figs. 7a, 8b). The labeling with deuterium in ʟ-dopa-d3 allowed us to track the distribution and metabolism of ʟ-dopa-d3 to dopamine-d3 in the brain and gut (Fig. 7).

First, we also investigated the effect of Mito-*ortho*-HNK on ʟ-dopa metabolism to dopamine in the gut[9]. Results show that Mito-*ortho*-HNK inhibits the conversion of ʟ-dopa-d3 to dopamine-d3 in the gut (Fig. 7b). Mice treated with ʟ-dopa-d3 (150 mg/kg) or ʟ-dopa-d3 (150 mg/kg) + Mito-*ortho*-HNK (10 mg/kg) by oral gavage show ʟ-dopa metabolism and dopamine-d3 formation by gut microbiome from gut-homogenized samples (Fig. 7a, b). Mito-*ortho*-HNK inhibited dopamine-d3 formation (Fig. 7b) and ʟ-dopa-d3 consumption (Fig. 7b) as compared with the control mouse in gut homogenized samples. In the initial proof-of-concept study in regular, wild type mice treated with ʟ-dopa-d3 or ʟ-dopa-d3 + Mito-*ortho*-HNK, a higher amount of dopamine-d3 was detected in brain samples in mice treated with ʟ-dopa-d3 + Mito-*ortho*-HNK compared with mice treated just with ʟ-dopa-d3 (Fig. 7c). This result indicates that Mito-*ortho*-HNK can inhibit ʟ-dopa-d3 consumption by gut microbiome, thus

increasing the amount of ʟ-dopa-d3 reaching the mouse brain and, consequently, increasing the formation of dopamine-d3 in the mouse brain.

## Discussion

Our scientific premise is that the benefits of ʟ-dopa treatment of PD in humans can be potentiated by the use of adjunctive treatments that inhibit ʟ-dopa breakdown in the gastrointestinal tract. The scientific impact is the strong potential of MTDs as nontoxic, and highly effective agents in mitigating gut metabolism of L-dopa to dopamine and enhancing the uptake and metabolism of ʟ-dopa into the brain in murine models of PD.

Studies have reported both beneficial and deleterious effects of antibiotics in the treatment of neurological disorders, including in PD[21]. Tetracycline antibiotics such as doxycycline and minocycline were reported to be effective in mitigating the progression of neuronal dysfunction in mice models of PD[22,23]. Neuroprotective mechanisms of antibiotics are distinctly different from their antibacterial mechanisms. Conventional antibiotics inhibit the growth of bacteria through inhibition of bacterial proteins, cell membranes, cell walls, or nucleic acid syntheses. In contrast, mitochondria-targeted agents exert cytostatic and not cytotoxic effects in E. *faecalis*. Studies report the identification of drug-resistant phenotypes and genes resistant to antibiotics in E. *faecalis*[24]. E. *faecalis* isolates displayed a high level of resistance to several traditional antibiotics including tetracycline antibiotics[25].

Mitochondria-targeted drugs may have some advantages over the traditional antimicrobials. Mitochondria-targeted drugs are more potent than the conventional antibiotics in inhibiting the proliferation of E. *faecalis* (Table 1, Fig. 4). Although we have not yet examined other commensal species expected to be present in the gut, our results suggest that the dose-dependence of this drug will likely differ for different species and different locations within the gut. However, our present findings indicate that the impact of mitochondria-targeted drugs (e.g., Mito-*ortho*-HNK and Mito-$PEG_5$-ATO) is temporary and may provide a window of opportunity to enhance the therapeutic efficacy of ʟ-dopa. Mito-*ortho*-HNK at 4

**Fig. 7 | Effect of Mito-*ortho*-HNK on ʟ-dopa/dopamine metabolism in vivo.** Mice were orally treated with 150 mg/kg ʟ-dopa-d3 with or without Mito-*ortho*-HNK for 2 h. The effects of Mito-*ortho*-HNK on dopamine-d3 formation and ʟ-dopa-d3 consumption were measured. ʟ-dopa-d3 and dopamine-d3 were separated and monitored by LC-MS-SIM. Standards are shown in (**a**). Effects of Mito-*ortho*-HNK on ʟ-dopa-d3 consumption and dopamine-d3 formation in the mice gut samples were measured, and the representative LC-MS-SIM trace are shown in (**b**). The formation of dopamine-d3 in mice brain samples was measured as described in the Methods section. **c** The amount of dopamine-d3 in the brain is based on the area under the curve of the LC-MS-SIM dopamine-d3 peak. Statistical significance was determined using a Student's *t* test. The values shown are the fold change in Mito-*ortho*-HNK-treated mice compared with controls. (\*\*) Statistically significant increase in dopamine-d3 formation compared with control (fold change>15 and *P* < 0.01). The representative LC-MS-SIM traces are shown in (**d**).

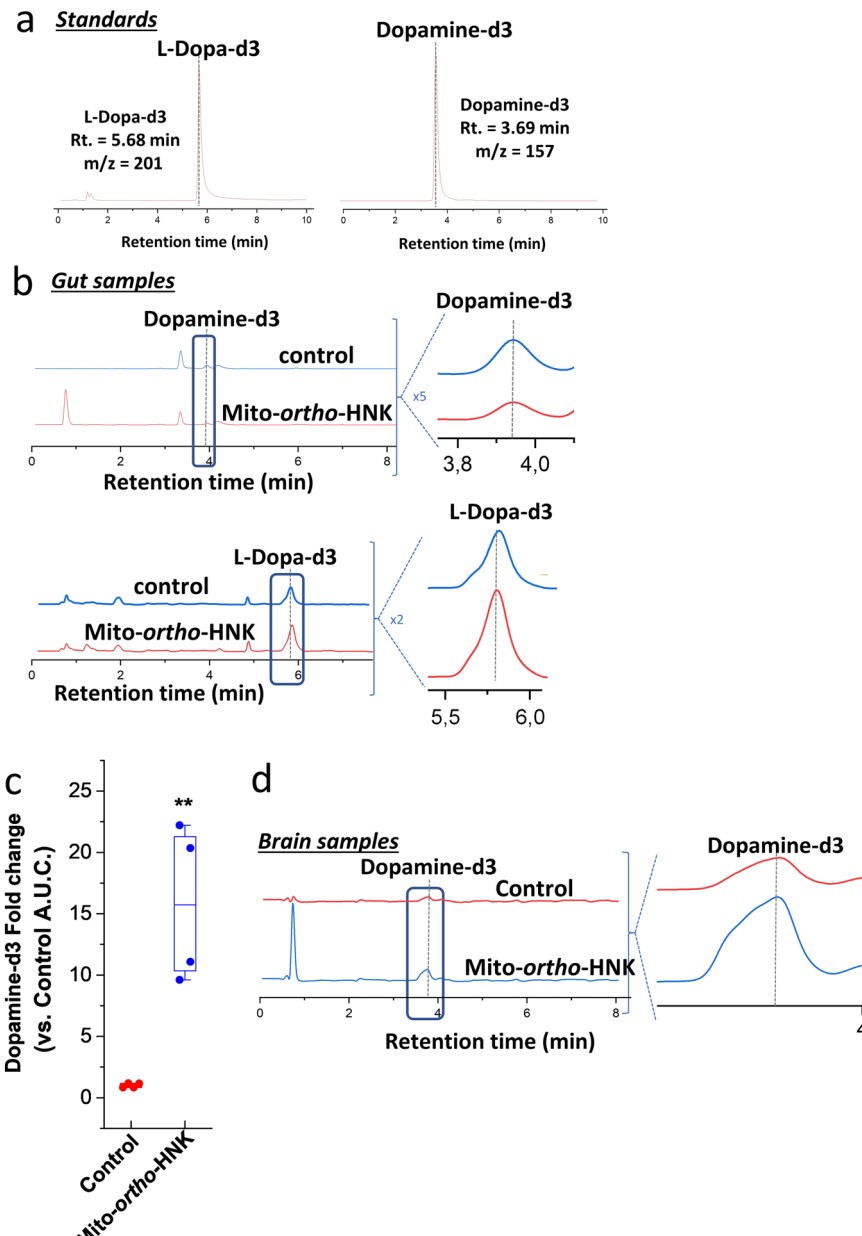

micromolar levels caused a substantial decrease in membrane potential in *E. faecalis* (Fig. 5a). At this concentration, the proliferation of *E. faecalis* was inhibited (>90%) for 2–3 h (Fig. 5b). The SYTOX assay indicated that Mito-*ortho*-HNK was not cytotoxic at this concentration (Fig. 5b). Previously, it has been reported that bacterial cell division is dependent on membrane potential[26]. The antibacterial effect of Mito-*ortho*-HNK is mechanistically different from those of chloramphenicol and ampicillin. Both chloramphenicol and ampicillin inhibited *E. faecalis* proliferation but did not affect its membrane potential (Fig. 5a). However, ampicillin inhibits bacterial cell wall synthesis, making the bacterial envelope more fragile, and did not affect membrane permeability (Fig. 5). Chloramphenicol inhibits bacterial protein synthesis and did not affect the membrane permeability. Consistent with these findings, ampicillin enhanced the uptake of SYTOX Green dye and chloramphenicol had little or no effect on SYTOX Green uptake (Fig. 5b). At these concentrations, the proliferation of *E. faecalis* was inhibited (Fig. 5a).

The antimicrobial efficacy of MTDs varies depending on the bacterial strain (Table 1). Most gram-positive bacteria were sensitive to MTDs, with MICs in the low micromolar to submicromolar range. However, MTDs exhibited less antibacterial efficacy toward *E. coli* and *P. aeruginosa* (Table 1, Fig. 4). Previous studies suggest that the presence of the highly effective multidrug resistance pump in *E. coli* is the reason why SkQ1, a decyl(triphenyl)phosphonium cation, conjugated to a quinone moiety was much less sensitive in *E. coli* (Table 1, Fig. 4). *E. coli* mutants lacking this resistance pump showed increased sensitivity to MTDs[16]. The multidrug resistance pump expelled the TPP⁺-containing SkQ1[16].

In a previous study, we reported the lack of in vivo toxicity of Mito-HNK[18]. The potential toxicity and neurological changes induced by Mito-HNK were assessed in an eight-week toxicology study in A/J mice[18]. Mice were treated with vehicle control and with various doses of Mito-HNK (7.5, 37.5, and 75 μmol/kg, which represent 2×, 10×, and 20× the therapeutically effective dose of 3.75 μmol/kg in the lung cancer mouse model[18], respectively), given via oral gavage five days per week for eight weeks. After eight weeks of treatment, no meaningful differences were observed between control mice and those treated with any dose of Mito-HNK, including the highest dose (75 μmol/kg)[18]. In addition, no histopathological changes were

**Fig. 8 | Gut microbial metabolism. a** Shows the gut metabolism of L-dopa and *m*-tyramine, and (**b**) shows MTD inhibition of gut bacterial metabolism of L-dopa to dopamine and *m*-tyramine.

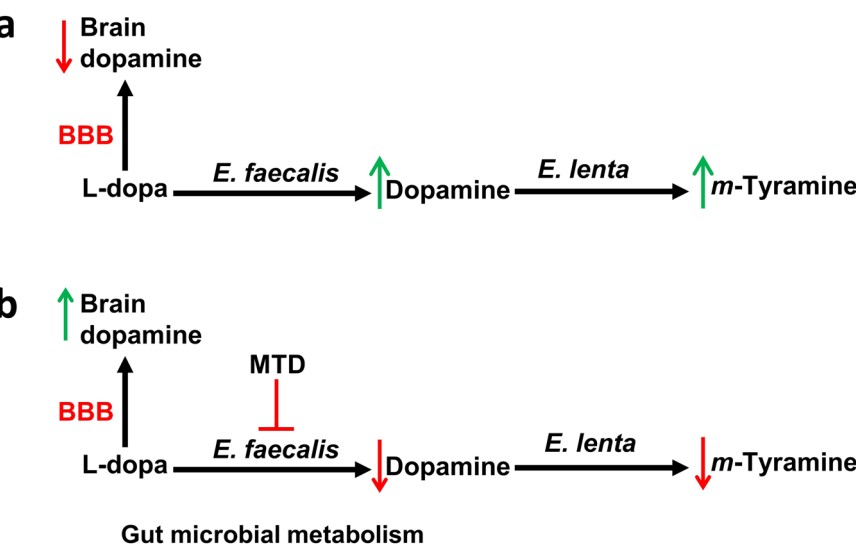

seen in either neural (frontal cortex and cerebellum) or skeletal muscles[18]. Mito-HNK did not affect the motor function monitored in a rotarod assay[18]. Overall, Mito-HNK did not show any indications of toxicity at a dose that is 20-fold higher than the maximally tolerated dose[18].

PEGylation decreases the hydrophobicity of MTDs, and nearly all MTDs and Mito-PEG analogs have similar alkyl side-chain lengths but substantially different hydrophobicities (log P [octanol partition coefficients]). The calculated log P values for atovaquone, HNK, lonidamine, Mito$_{10}$-ATO, Mito-*ortho*-HNK, Mito-lonidamine, Mito-PEG$_5$-ATO, Mito-PEG$_4$-HNK, and Mito-PEG$_4$-APO are 5.1, 5.2, 4.5, 12.8, 13.0, 9.3, 9.2, 9.5, and 5.4, respectively (Supplementary Table 1). This approach enables us to vary drug hydrophobicity and enhance the antimicrobial effects of MTDs.

Previous studies using another form of deuterated L-dopa, SD-1077 (where the alkyl side chain protons are deuterated), indicate a slower metabolism of the corresponding deuterated dopamine (due to the kinetic isotope effect), leading to enhanced stability and behavioral potency in an animal model of PD[27,28]. The deuterated L-dopa used in this study has aromatic ring protons that are deuterated and, therefore, do not have the kinetic isotope effect. The endogenous levels of dopamine in the mouse brain are relatively high, and the increased dopamine levels in the brain that result from use of the nondeuterated conventional form of L-dopa might make optimization of the method more difficult. The use of appropriately labeled L-dopa with the carbon-13-labeled isotope should help overcome the detection limitations (of dopamine) with the possibility of method optimization.

A genetically engineered mouse model, the MitoPark mouse, recapitulates many of the phenotypic features (mitochondrial dysfunction, microglial activation, dopaminergic degeneration, dopamine deficiency, and progressive neuronal deficits and protein occlusion) of PD. Our scientific premise is that the benefits of treating human PD with L-dopa can be potentiated by the use of adjunctive treatments utilizing MTDs that inhibit the breakdown of L-dopa to dopamine in the gastrointestinal tract, thereby enhancing L-dopa conversion to dopamine in the brain (Fig. 8). Future studies focusing on mouse models of PD, including MitoPark and LRRK2 transgenic mice, should provide additional insights relevant to gut microbial metabolism of L-dopa and its inhibition by Mito-*ortho*-HNK. Furthermore, MTDs (e.g., Mito-apocynin and Mito-quinone) prevent hyposmia and loss of motor function in LRRK2 PD mice, and they inhibit MPTP (1-methyl-4-phenyl-1,2,3,6-tetrahydropyridine)-induced neurotoxicity in a PD mouse model[29,30]. A combination of L-dopa and carbidopa is the treatment of choice for managing PD symptoms. Carbidopa does not prevent gut metabolism of L-dopa but does inhibit the peripheral metabolism of L-dopa by acting as a substrate inhibitor of peripheral amino carboxylases. MTDs could enhance the efficacy of L-dopa/carbidopa therapy by directly reversing the gut bacteria metabolism of L-dopa. Thus, it is conceivable that a combination of MTD/L-dopa/carbidopa therapy may inhibit both the microbial metabolism and the peripheral metabolism of L-dopa.

## Methods
### Syntheses of MTDs
MTDs (e.g., Mito-*ortho*-HNK) were prepared by conjugating the mitochondria-targeting TPP$^+$ moiety to the corresponding parent molecule via alkyl linkers of different natures and lengths (Fig. 9). Based on the data published by us and other laboratories[31–35], fine-tuning of the linker length and hydrophobicity is important for optimization of the uptake and biological activity of the TPP$^+$-based MTDs (Supplementary Table 1).

To a mixture of HNK (1.3 g, 4.9 mmol), anhydrous potassium carbonate (0.69 g, 4.9 mmol) in DMF (40 mL) was added 10-bromodecyltriphenylphosphonium bromide (2.8 g, 4.9 mmol). The mixture was stirred at 40 °C for 24 h. The solvent was removed under vacuum, and the residue was taken up into water and extracted with dichloromethane. The organic layer was dried over sodium sulfate, and the solvent was removed under reduced pressure. Purification by flash chromatography (diethyl ether, dichloromethane, and dichloromethane/ethanol, 9/1) delivered the corresponding mitochondria-targeted *ortho*-honokiol (Mito-*ortho*-HNK) (0.3 g, 8.1% yield). Additional details are provided in the Supplementary Information (Supplementary Fig. 1-3 and Supplementary Table 1).

### Bacterial proliferation assay
All in vitro experiments were conducted using validated bacterial strains acquired from the American Type Culture Collection for *E. faecalis* (Cat# OG1RF) and from the National Collection of Type Cultures for *Eggerthella lenta* (Cat# NCTC 11813). *E. faecalis* was cultured in tryptic soy broth. *E. faecalis* cells were diluted 1:100 into fresh tryptic soy broth medium. Then, they were grown at 37 °C in flasks on a rotating shaker at 250 rpm to reach the exponential growth phase (optical density at 600 nm [OD$_{600}$] of 0.2–0.5) before use in the in vitro assays. For all proliferation assays, cells were diluted to the final OD$_{600}$ of 0.1 with indicated treatments in a 96-well plate. Cell proliferation, which was represented as absorbance at 600 nm, was acquired in real time every 3 min for 6 h using a plate reader (BMG Labtech, Inc., Ortenberg, Germany) equipped with an atmosphere controller set at 37 °C, 100% air.

### Bacterial metabolism of L-dopa
*E. faecalis* cells in the exponential growth phase (OD$_{600}$ of ~0.4) were diluted to the final OD$_{600}$ at 0.1. Then, they were treated in the presence of L-dopa

**Fig. 9 | Chemical structures of drugs, MTDs and PEGylated MTDs, and L-dopa and analogs.** Chemical structures of drugs, MTDs, and PEGylated MTDs are shown in (**a**), and of L-dopa and analogs are shown in (**b**).

(1 mM) alone or in combination with carbidopa (0.22 mM) in the same manner as described for Mito-*ortho*-HNK. At the indicated time points (1–6 h), samples (1 ml of media) were collected by centrifugation at 2500 $g \times$ 5 min at 4 °C and stored at −80 °C before performing lyophilization. The dry residue was dissolved in ice-cold methanol (100 µl) and taken for liquid chromatography–mass spectrometry (LC-MS) analysis.

### LC-MS experiments
L-Dopa and its metabolites were separated and monitored by LC-MS using an Agilent 1200 apparatus equipped with an ultraviolet-visible absorption and mass spectrometry detector (single quadrupole). Typically, 2 µL of a sample was injected into an Agilent Poroshell column (120 HILIC-Z, PEEK, 100 mm × 2.1 mm, 2.7 µm, 25 °C). The absorption traces were collected at 280 nm.

L-Dopa-d3 and dopamine-d3 were separated and monitored by liquid chromatography–mass spectrometry–single ion monitoring (LC-MS-SIM) using an Agilent 1200 apparatus equipped with an ultraviolet-visible absorption and MS detector (single quadrupole). Typically, 2 µL of a sample was injected into an Agilent Poroshell column (120 HILIC-Z, PEEK,

100 mm × 2.1 mm, 2.7 µm, 25 °C) equilibrated with 100% ammonium formate (10 mM, pH 3.0 containing acetonitrile/water, 9/1). The compounds were separated by a linear increase in ammonium formate (10 mM, pH 3.0 containing acetonitrile/water, v/v) phase concentration from 0% to 80% over 14 min using a flow rate of 0.5 mL/min. The absorption traces were collected at 280 nm.

In addition, mass spectrometry–single ion monitoring (MS-SIM) detection parameters were set up using electrospray ionization. SIM was defined as follows: L-Dopa-d3 [$m/z = 201$ (+)] and dopamine-d3 [$m/z = 157$ (+)]. Additional details are provided in the Supplementary Information.

### Bacterial membrane potential and cytotoxicity assay
The effects of MTDs on bacterial membrane potential were determined using the fluorescent dye TMRM[12–14]. Briefly, bacteria in the exponential growth phase ($OD_{600}$ of 0.4) were treated as indicated for the MTDs or commonly used antibiotics in a black, clear-bottom 96-well plate; then, an aliquot of TMRM was added at a final concentration of 50 nM for 20 min[6,12,14,36]. After incubation with TMRM, the plate was centrifuged twice

at 2500 $g$ for 5 min and washed with phosphate-buffered saline. Fluorescence was monitored at an excitation of 544 nm and emission of 590 nm using a plate reader (BMG Labtech, Inc.). Data were collected as the mean fluorescent intensity. Results were normalized to the total $OD_{600}$ as the total bacteria number.

For kinetic monitoring of the cytotoxicity assay, *E. faecalis* cells in the exponential growth phase ($OD_{600} = 0.4$) were treated as indicated for the MTDs or commonly used antibiotics in a black, clear-bottom 96-well plate for 3 h, and dead cells were monitored in the presence of 200 nM SYTOX Green (Invitrogen, Waltham, MA). The SYTOX method labels the nuclei of dead cells, yielding green fluorescence. Fluorescence intensities from the dead cells in the 96-well plate were acquired in real time every 5 min for 3 h using a plate reader (BMG Labtech, Inc.) equipped with an atmosphere controller set at 37 °C, 100% air using a fluorescence detection with 485 nm excitation and 535 nm emission. Data are represented as mean fluorescent intensity.

## Intracellular ATP levels

A luciferase-based assay was used to measure intracellular adenosine triphosphate (ATP) levels according to the manufacturer's instructions (Sigma Aldrich, St. Louis, MO, Cat# FLLAA). Briefly, a mixture containing luciferase and luciferin (Cat# FLAAM) was added to cell lysates. After swirling, the light released was measured in a luminometer. The results were normalized to $OD_{600}$ level in each well.

## In vivo detection of dopamine-d3

All animal protocols were approved by the Medical College of Wisconsin Institutional Animal Care and Use Committee. Mice (C57BL/6J, 6–8 weeks old, from Jackson Laboratories) were divided into two groups and then orally gavaged with either 250 mg/kg L-dopa-d3 only (control) or 250 mg/kg L-dopa-d3 + 20 mg/kg Mito-*ortho*-HNK for 2 h. The mice were sacrificed, and the brain tissue and gut tissue (including stomachs and intestines) were harvested, snap frozen in liquid nitrogen, and then stored at −80 °C before extraction. The protocol for extracting L-dopa and dopamine is the same as described previously[37]. Once tissue weight was obtained, it was transferred to a homogenization tube containing 1 ml of ice-cold methanol. The tissue homogenization and extraction were performed using an Omni Bead Ruptor 24 homogenizer (Omni International, Kennesaw, GA). The homogenate was centrifuged at 16,000×g for 10 min 4 °C; then, the supernatant was collected and transferred to a new 1.5 ml microcentrifuge tube for lyophilization. All samples were stored at −80 °C until LC-MS analysis. L-dopa-d3 and dopamine-d3 were separated and monitored by LC-MS-SIM. The absorption traces were collected at 280 nm, and MS-SIM detection parameters were set up using electrospray ionization. SIM was defined as follows: L-dopa-d3 [$m/z = 201$ (+)] and dopamine-d3 [$m/z = 157$ (+)].

## Statistics and reproducibility

Comparisons between the control and treatment groups were made using an unpaired Student's $t$ test analysis. Sample size and replicates are stated in figure legends, respectively. $P$-values of less than 0.05 were determined as statistically significant. Values denote mean ± standard deviation (SD) or mean ± standard error of the mean (SEM). The number of replicates per treatment group are shown as $n$.

## Data availability

All data generated or analyzed during this study are included in this article and the accompanying supplementary files. The source data underlying all graphs in the manuscript are provided in Supplementary Data 1.

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

## Acknowledgements

Research reported in this publication was supported in part by the National Cancer Institute of the National Institutes of Health under Award Number R01CA208648 (B.K.), the National Cancer Institute of the National Institutes of Health under Federal Award RS 20182223-00 (B.K.), a program project formation grant from the Advancing a Healthier Wisconsin Endowment (C.H.), the G. Frederick Kasten, Jr Endowed Professorship in Parkinson's Disease Research (C.H.), and the Harry R. and Angeline E. Quadracci Professor in Parkinson's Research Endowment (B.K.), and International Research Project SuperO$_2$ from CNRS, France (M.H.). The content is solely the responsibility of the authors and does not necessarily represent the official views of the National Institutes of Health. Thanks to Lydia Washechek for preparing and proofreading the manuscript.

## Author contributions

Conceptualization, B.K., J.F., C.H., M.H., and G.C.; methodology, B.K., J.F., M.H., C.H., G.C.; software, M.H. and G.C.; validation, B.K., J.F., C.H., M.H., and G.C.; formal analysis, B.K., G.C., J.F., M.H.; investigation, G.C., and M.H. and J.F.; resources, B.K., J.F., C.H., and M.H.; data curation, G.C. and M.H.; writing—original draft preparation, B.K.; writing—review and editing, B.K., J.F., C. H., G.C., and M.H.; visualization, G.C., and M.H.; supervision, B.K. and J.F.; project administration, B.K., J.F., C.H., and M.H.; funding acquisition, B.K., J. F., C. H., and M.H. All authors have read and agreed to the published version of the manuscript.

## Competing interests

Balaraman Kalyanaraman and Micael Hardy are inventors of US Patent No. 10,836,782/European Patent No. 3307254, "Mito-honokiol compounds and methods of synthesis and use thereof." The other authors have no competing interests to declare.
