## [Peer review file · Communications Biology]

Reviewers' comments:

Reviewer #1 (Remarks to the Author):

This is a very interesting study and a well-written paper. Premature metabolism of levodopa is a serious problem for many people with Parkinson's disease and the therapeutic compound researched in this study is a novel and interesting approach to combating this problem.

I have a few comments/questions I would like to see amended/answered in a further version of the paper:

1. Is Mito-ortho-HNK a proprietary name? If so, I would strongly recommend using chemical nomenclature throughout the article, with the brand name only mentioned once to aid replicability of the study.
2. Do I understand correctly that Mito-ortho-HNK inhibits bacterial proliferation as a whole, and not just that of *E. faecalis*? If so, are negative effects to be expected from inhibited growth of mutualistic/commensal symbionts? The paper could reflect this consideration.
3. An important reason for diminished and/or unpredictable efficacy of levodopa is gastroparesis, which is present in up to 100% of Parkinson's disease patients. The Mito-ortho-HNK administration, however, seems to be quite time-sensitive, given the exhibited time lag. How do the authors propose to overcome this impairment in humans?
4. The correct full name of AADC is "Aromatic L-amino acid decarboxylase", the authors mistakenly wrote "L-aromatic".
5. The authors mention dyskinesia as a side effect of increased dosages of levodopa. It is not. Rather, it is a result of altered neuronal dopamine sensitivity which comes with advanced disease (the so-called 'ON-OFF phenomenon'). E.g. the LEAP study showed that even drug-naive patients develop dyskinesias at low doses of levodopa as long as their illness is advanced enough.
6. I feel that "the increased presence of gut bacteria in PD patients" is worded imprecisely. Gut bacteria as a whole are not 'increased' in PD, to my knowledge. Rather, the composition of the microbiome is different from that in non-PD individuals and the gut bacteria may be more abundant in places where they normally do not grow in such numbers (such as in small-intestinal bacterial overgrowth, SIBO).
7. The authors claim that "these compounds have previously been used in preclinical animal models of PD and in patients with PD without significant toxicity" and give reference 18 to back up this claim. However, reference 18 does not include any reference to Parkinson's disease, as far as I can see. A new reference is needed.

Reviewer #2 (Remarks to the Author):

This paper addresses the topic of L dopa metabolism in Parkinson's disease. The connection between gut microbiome and Parkinson's disease is an important and rapidly growing topic and research period the gut microbiome is also increasingly recognized as a potential source of variability in how Parkinson's medications are metabolized, as well as their bioavailability and pharmacokinetics. These associations are highly relevant to real world patients and may guide clinical management strategies and novel therapeutics. Therefore the hypothesis and findings in this paper are foundationally well supported by current knowledge and will be of interest to researchers

The main findings of this paper are that mitochondrial targeted drugs - with a focus on a particular Mito ortho HNK compound - inhibit activity of *E. Faecalis* thus decreasing the bacterial metabolism of L dopa and allowing more L dopa to reach the brain to be converted into dopamine. These findings are rigorously supported and have high translational potential. The authors present a variety of experiments that highlight the various properties and effects of the MiTo ortho HNK as well as other related compounds, assess their potential for toxicity, and compare them to traditional antibiotics like ampicillin.

Some strengths of this paper are the experimental approach, clarity of the findings, and translational potential.

Some weaknesses include relative lack of clarity on the rationale or justification for the selection of the mitochondrial targeted drugs that are focused on here. Additionally the focus is on a single bacteria family. The background sections would benefit from increased detailed explanation synthesizing why this particular bacterial family is of prime relevance and how it was selected based on the studies cited and others. The discussion section would benefit from additional emphasis on why the proposed mitochondrial targeted drugs could have relative advantages over traditional antimicrobials. One final relatively minor detail is that reference 16 appears to be cited in error and should be replaced with a correct citation.

Response to Reviewers

We thank the reviewers for the constructive comments. We have revised the manuscript accordingly and, as a result, we feel the manuscript is very much improved.

Reviewer #1

1. Is Mito-ortho-HNK a proprietary name? If so, I would strongly recommend using chemical nomenclature throughout the article, with the brand name only mentioned once to aid replicability of the study.

Response:

Mito-ortho-HNK is not a proprietary name. It is a chemical name for one of the isomers of Mito-HNK. The structure of this compound is shown in Fig. 1.

2. Do I understand correctly that Mito-ortho-HNK inhibits bacterial proliferation as a whole, and not just that of E. faecalis? If so, are negative effects to be expected from inhibited growth of mutualistic/commensal symbionts? The paper could reflect this consideration.

Response:

As shown in Figure 5 and Table 1, Mito-ortho-HNK has variable effects on several bacterial strains, with efficacy primarily against Gram positive organisms among those strains tested. We have not yet examined other commensal species expected to be present in the gut. Based on our current results, those effects may be expected to be complex (with dose dependence differing for different species and different locations within the gut). A broad survey of the effects of Mito-ortho-HNK against common commensal bacteria is well beyond the scope of this paper. However, the point we make is that the impact of Mito-ortho-HNK is temporary, such that a long-term negative impact is unlikely (compared, for example, with current clinically used antibiotics).

3. An important reason for diminished and/or unpredictable efficacy of levodopa is gastroparesis, which is present in up to 100% of Parkinson's disease patients. The Mito-ortho-HNK administration, however, seems to be quite time-sensitive, given the exhibited time lag. How do the authors propose to overcome this impairment in humans?

Response: At present, we don't know whether Mito-ortho-HNK could help overcome the gastroparesis impairment. However, published reports suggest that polyphenols exert gastroprotective effects against gastro-intestinal disorders (Chiu H-F et al. *Molecules* 26, 2090, 2021; Zhang W et al. *Front Immunol* 12, 620510, 2021) (lines 224-226). The reviewer raises an important point and additional research using mitochondria-targeted polyphenolic compounds like Mito-ortho-HNK may provide new insights into this important medical issue.

4. The correct full name of AADC is "Aromatic L-amino acid decarboxylase", the authors mistakenly wrote "L-aromatic".

Response: This has been corrected as suggested by this reviewer (line 39).

5. *The authors mention dyskinesia as a side effect of increased dosages of levodopa. It is not. Rather, it is a result of altered neuronal dopamine sensitivity which comes with advanced disease (the so-called 'ON-OFF phenomenon'). E.g. the LEAP study showed that even drug-naive patients develop dyskinesias at low doses of levodopa as long as their illness is advanced enough.*

Response: While dyskinesias can develop even in drug-naive patients, the literature strongly supports that the incidence of LID is enhanced with longer treatment and increased L-dopa dosage. As per reviewer's suggestion, we have modified the manuscript to clarify this point (lines 43-44).

6. *I feel that "the increased presence of gut bacteria in PD patients" is worded imprecisely. Gut bacteria as a whole are not 'increased' in PD, to my knowledge. Rather, the composition of the microbiome is different from that in non-PD individuals and the gut bacteria may be more abundant in places where they normally do not grow in such numbers (such as in small-intestinal bacterial overgrowth, SIBO).*

Response: We agree with the reviewer. We corrected the wording appropriately in the revised manuscript (lines 47-52, 53-55).

7. *The authors claim that "these compounds have previously been used in preclinical animal models of PD and in patients with PD without significant toxicity" and give reference 18 to back up this claim. However, reference 18 does not include any reference to Parkinson's disease, as far as I can see. A new reference is needed.*

Response: Mito-ortho-HNK has not been tested in a preclinical animal model of PD. Reference 18 refers to another triphenylphosphonium-based compound (*i.e.*, Mito-apocynin). However, Mito-HNK (consisting of two isomers) was tested in a sub-chronic mice model (the new ref. 18) and showed a lack of toxicity. We changed the reference as suggested (line 88).

Reviewer #2

1. *...relative lack of clarity on the rationale or justification for the selection of the mitochondrial targeted drugs that are focused on here.*

Response: We have included the justification for selecting the triphenylphosphonium-based compounds used in this study (lines 81-84). The rationale for this study is based on the previously reported antibacterial effect of Mito-Q (Ref. 16).

2. ...the focus is on a single bacteria family.

Response: We have tested the efficacy of Mito-ortho-HNK and other analogs in 10 different bacteria from both gram positive and gram negative bacterial families. We focused on the gram positive *E. faecalis* (line 70).

3. The background sections would benefit from increased detailed explanation synthesizing why this particular bacterial family is of prime relevance and how it was selected based on the studies cited and others.

Response: We have expanded the background section as suggested by the reviewer indicating the relevance of this particular bacteria in understanding its role in the gut metabolism of L-dopa leading to decreased production of dopamine in the brain (lines 218-227).

4. The discussion section would benefit from additional emphasis on why the proposed mitochondrial targeted drugs could have relative advantages over traditional antimicrobials.

Response: As suggested by this reviewer, we have included in the discussion the advantages of TPP⁺-based mitochondria-targeted compounds over conventional antimicrobial antibiotics (lines 229-236).

5. One final relatively minor detail is that reference 16 appears to be cited in error and should be replaced with a correct citation.

Response: We have corrected this reference (line 81).

REVIEWERS' COMMENTS:

Reviewer #1 (Remarks to the Author):

I am satisfied by the authors' responses and alterations to the manuscript. I would support publication of the manuscript in its present form.
Line 61 contains a small typo (TryDC instead of TyrDC).

Reviewer #2 (Remarks to the Author):

The revised version of this manuscript adequately addresses the reviewer critiques. The justification is improved, and explanation of clinical relevance more appropriately highlighted. This manuscript adds novel and interesting new data to the field. I recommend accepting this paper for publication.

Response to Reviewers

We thank both reviewers for their careful review of the manuscript.

Reviewer #1

Comment: Line 61 contains a small typo (TryDC instead of TyrDC).

Response: Thank you. We have corrected the typo.